

# Climatological occurrences of hail and tornado associated with mesoscale convective systems in the United States

Jingyu Wang[1,2*], Jiwen Fan[1*], Zhe Feng[1]

[1]Pacific Northwest National Laboratory, Richland, WA 99354, USA

[2]National Institute of Education, Nanyang Technological University, Singapore 637616, Singapore

*Correspondence to*: Jiwen Fan (jiwen.fan@pnnl.gov) Jingyu Wang (jingyu.wang@nie.edu.sg)

**Abstract.** Hail and tornadoes are hazardous weather events that are responsible for significant property damage and economic loss world-wide. The most devastating hail and tornado events are commonly produced by supercells in the United States. However, some hazard-producing supercells may grow upscale into mesoscale convective systems (MCSs) or may be

embedded in MCSs. Quantifying the relationship of hail and tornado occurrences with MCSs on the long-term climatology is lacking. In this work, the radar features associated with MCSs are extracted from a 14-year MCS tracking database over the contiguous United States, and the hazard reports are matched to the extracted MCS features. We analyze the characteristics of hail and tornadoes associated with MCS characteristics and consider the seasonal and regional variabilities. On average, about 8-17% of hail and 17-32% of tornado events are associated with MCSs depending on various criteria used to define

MCSs. The maximum total and MCS-associated hazard events occur in March-May, but the highest MCS-associated portion (23% for hail and 45% for tornado) occurs in winter (December-February) because MCS is the dominant type of convection due to strong synoptic forcing. In contrast to the decreasing trend in the relationship of MCS-associated fraction with hailstone size, the more severe the tornado event is, the more likely it is associated with an MCS. The different trends indicate the distinct mechanisms for the MCS-associated production of the two types of hazards.

**1 Introduction**

Severe convective storms produce a variety of hazards including hail, tornadoes, and strong winds, which threaten human lives and cause tens of billions of dollars in economic losses globally each year (e.g., Munich, 2016). In the United States, the total economic losses from severe convective storms were $125 billion during 2007–2016, outweighing all other natural hazards including flooding, droughts, and tropical cyclones (Baggett et al. 2018). Hail and tornadoes are highly localized and

short-lived in nature, which poses a significant challenge in their prediction.

Hail can be produced by a variety of convective organizations (Trapp et al. 2005; Gallus et al. 2008; Smith et al. 2012), commonly associated with supercell thunderstorms (Moller et al., 1994), but is also found in multicellular storms (Nelson, 1987), and pulse storms (isolate, short-lived thunderstorms forming in a weakly forced environment; Miller and Mote, 2017). Much like hail, tornadogenesis is strongly associated with supercells (Schumacher and Rasmussen, 2020). As a result,


supercell is typically considered the most prolific producers of hail and tornadoes relative to other modes of convection, such as ordinary cells, multi-cells, mesoscale convective systems (MCSs), and unclassifiable types of convection (Duda and Gallus, 2010). Past studies on the relationship of hail and tornado with MCSs focused on quasi-linear convective systems (QLCSs; a subset of MCSs). Derived from a 22-year (1996-2017) observational dataset, 10% of severe hail (with hailstone diameter greater than or equal to 1 inch) and 21% of tornadoes were attributed to QLCSs over the entire contiguous United

States (CONUS) by Ashley et al. (2019). Using data from a shorter period (2003-2011), Smith et al. (2012) reported that 0.8% of significant severe hail (with hailstone diameter greater than or equal to 2 inches) and 13.8% of tornadoes were produced by QLCSs. For significant tornadoes (with the Enhanced Fujita scale [EF] rating of 2-5), 20% of those events were linked to QLCSs from 2000 to 2008 (Grams et al., 2012). For the most extreme case, a QLCS has been reported for the unprecedented production of 76 tornadoes on 27 April 2011 (Knupp et al., 2014).

The environments that support hail-producing storms show strong and persistent updrafts (Blair et al., 2017), large convective available potential energy (CAPE), 0-6 km bulk wind shear (Craven et al., 2004), as well as 0-3 km storm-relative helicity (Prein and Holland, 2018). Compared to the formation of hail, tornadogenesis may have different environmental conditions including high surface moisture supply and directional wind shear in the lowest 1-km (Rasmussen and Blanchard, 1998; Rasmussen, 2003). Thus, most of the tornado occurrences are supercell-related (Smith et al., 2012),

accounting for over 90% of tornado deaths in the contiguous United States (CONUS, Schoen and Ashley 2011; Brotzge et al. 2013). However, such environments ideal for hail production and tornadogenesis are not only found in supercells exclusively, some MCSs also develop in similar environments as revealed by observations (Nielsen and Schumacher, 2020) and numerical simulations (Nielsen and Schumacher, 2018), such as moist boundary layer with large, low-level vertical shear, therefore supercell-like rotate can be embedded (Schumacher and Rasmussen, 2020).

MCSs are considered the largest form of convective clouds (Houze, 2004) with spatial scales of 100s km and a lifespan of several hours to beyond a day (Houze, 2015; Feng et al., 2018, 2019). MCSs are commonly observed in many regions of the Earth (Wang et al., 2019a; Houze et al., 2019; Feng et al. 2021) and significantly impact precipitation, radiative forcing, and general circulation (Houze, 2018). Hazard-producing supercells, multi-cells and pulse storms can grow upscale and merge with nearby storms into an MCS. In addition, new quasi-linear or clustered convective features can be generated near pre-

existing boundaries such as fronts or outflow boundaries, convectively generated gravity waves, or mesoscale convective vortices (Schumacher, 2009) within MCSs, and subsequently produce hail and tornado.

    Quantifying the relationship of hail and tornado occurrences with MCSs which have large spatial (over 100 km) and temporal scales (over 10 hours) has important implications in understanding the predictability of weather hazards at the medium-range forecasts (i.e., subseasonal to seasonal scales). Many recent studies have empirically related the change of

hazard activities and their spatial pattern over the U.S. to the large-scale disturbances, such as the Global Wind Oscillation (Gensini and Allen, 2018; Gensini and Marinaro, 2016; Moore, 2017), El Niño/Southern Oscillation (ENSO; Allen et al., 2015; Childs et al., 2018), and the Madden-Julian Oscillation (MJO; Barrett and Gensini, 2013; Barrett and Henley, 2015; Baggett et al., 2017; Baggett et al., 2018). More recently, the environmental factors most likely contributing to the



interannual variabilities of hail occurrence over southern and northern Great Plains have been investigated by Jeong et al.
(2020 and 2021), suggesting a statistical method of hail prediction. The direct empirical linkages between hazard activities
and large-scale features indeed have a practical implication for the seasonal predictability of such events (e.g., Lepore et al.,
2017). However, there is an important scale gap that needs to be filled to better understand how large-scale features could
regulate the hazard activities at the micro-mesoscale. With MCSs existing at the mesoscale, we can not only gain an
improved understanding of the mechanism of how the large-scale features influence the local hazard activity by establishing
MCS-hazard relationships, but also improve the medium-range (sub-seasonal) forecasts by adding mesoscale factors to their
prediction. However, a long-term climatological quantification of the relationships of hail and tornado with MCSs is lacking.
This study aims to quantify the relationships of hazard events with associated MCSs based on the long-term (2004-2017)
National Oceanic and Atmospheric Administration (NOAA) hazard reports and an MCS observational dataset over the
CONUS. By mapping the instantaneous hazard reports to the MCS dataset (Feng, 2019) and linking them to the convective
features embedded within the tracked MCSs based on the National Weather Service Next-Generation Radar (NEXRAD)
observations, we analyzed the characteristics of hail and tornadoes associated with MCS characteristics and quantify the
seasonal and regional variabilities.

This study is organized as follows: Section 2 provides an overview of the hazard report database and the MCS dataset, as
well as the methodology used to map the two datasets spatially and temporally. Section 3 shows the results of the statistical
characteristics of hail and tornado events and their relationships with MCS properties. Summary and discussion are
presented in Section 4.

## 2 Data and methodology

### 2.1 Hail and tornado data

The hail and tornado reports are obtained from the NOAA National Centers for Environmental Information (NCEI)'s Severe
Weather Data Inventory (SWDI), Storm Events Database (available at https://www.ncdc.noaa.gov/stormevents/). The SWDI
integrates severe weather reports over the U.S. from a variety of sources, which records the individual event as a data entry
with detailed time (in minutes), point-based geographic coordinates (with 0.001° precision), and magnitude (hail in inches
and tornado in Enhanced Fujita scale [EF] rating). Both hail and tornado reports are treated as point events with the start time
and location in this study. As mentioned by many previous studies (e.g., Kelly et al. 1985; Doswell and Burgess 1988; Weiss
et al. 2002; Doswell et al. 2005; Verbout et al. 2006; Trapp et al. 2006; Smith et al. 2006), the Storm Events Database is
greatly influenced by population density, and there exist uncertainties in the assignment of tornado intensity. However, as
noted in Jeong et al (2020 and 2021), with the guidance of NEXRAD, the hail and tornado report data have an improved
accuracy through verification of severe storm occurrences in recent two decades (Allen and Tippett, 2015). Thus, we are
focusing on the most recent decade (2004-2017) in this study.



## 2.2 MCS identification and tracking

MCSs are poorly identified and there is no consensus regarding the observational inputs and thresholds used. Some previous studies are solely based on the segmentation of radar imagery using thresholds of size, intensity, and spatial pattern built on extensive expert analyses (Smith et al., 2012; Parker and Johnson, 2000). Because of the intrinsic caveats of radar reflectivity data (e.g., insufficient radar overlap, excessive removal of anomalous propagation and false echoes), Erroneous MCS segmentation is commonly found in previous studies, including excessive, missing, and incorrect merging of MCS candidates (Haberlie and Ashley, 2018a). Motivated by those issues, a more sophisticated MCS segmentation and tracking algorithm was developed by Haberlie and Ashley (2018a, b), where an MCS candidate is detected if the major axis length of the aggregated convective cells (composite reflectivity ≥ 40 dBZ, aggregation radius of 6, 12, 24 or 48 km) exceed 100 km, and its boundary is defined as the contiguous stratiform precipitation (composite reflectivity ≥ 20 dBZ) within a certain search radius (48, 96, or 192 km) from the MCS candidate. After detection, multiple properties of the entire object will be examined for the likelihood to be a 'true MCS' using a tree-based machine learning technique. Finally, only the most likely object will be tracked through a spatiotemporal overlap method, and the ones that persist more than 3 h are defined as MCSs. In addition to the signature on radar imagery, MCSs also feature the large-coverage, long-lasting precipitation and the associated hydrological hazards of flood and flash flood (Hu et al. 2021; 2022). Focusing on precipitation features associated with MCSs, Feng et al. (2019) developed an MCS tracking dataset by applying the FLEXible object TRacKeR (FLEXTRKR) algorithm (Feng et al. 2018) on a long-term dataset that synthesized geostationary satellite and NEXRAD radar observations. The FLEXTRKR algorithm starts with tracking contiguous cold cloud systems (CCS) defined by contiguous areas with satellite infrared brightness (IR) temperature (Tb) < 241 K. If a tracked CCS meets the size (exceeding $6 \times 10^4$ km2) and duration (longer than 6 h) thresholds, its embedded convective features (using NEXRAD 3D radar reflectivity) and precipitation features (using Stage IV precipitation product) are further examined. If the largest precipitation feature with a major axis length > 100 km and embedded convective echoes exceed 45 dBZ, and these conditions persist for longer than 6 h, the tracked CCS is defined as an MCS, which includes all the radar echo objects under the anvil cloud shields.

Since FLEXTRKR uses the CCS boundary defined by IR temperature, the identified MCSs are very large in spatial scale (referred to as IR-MCSs), which might cover multiple convective systems. To be more consistent with the literature work on studying the MCS-hazard relationship which is based on radar features (Haberlie and Ashley, 2018a, b), we developed a methodology to further define MCS radar features (referred to as radar-MCSs) within tracked IR-MCSs based on radar reflectivity in the FLEXTRKR database, as detailed blow.

As illustrated in Fig. 1, we first examine the embedded convective cores defined by the Storm Labeling in Three Dimensions (SL3D) classification algorithm (Starzec et al., 2017) within IR-MCSs. Compared to the simple threshold-based method for identifying convective cores (e.g., composite reflectivity ≥ 40 dBZ with the embedded area of reflectivity ≥ 50 dBZ exceeding 40 km2, [Haberlie and Ashley, 2018a]), the convective cores defined by SL3D fully utilizes the three-dimensional



volumetric radar data by considering both the vertical extent and horizontal gradient of reflectivity, thus the results better agree with convective updrafts (Starzec et al., 2017) that are more likely to produce hazardous weather. Meanwhile, the

contiguous reflectivity objects (i.e., the combination of convective and stratiform echoes defined by SL3D) are marked as the possible maximum radar-MCS boundary. In the second step, the convective cores within the IR-MCS shield are aggregated into a contiguous region of convection using a convective aggregation radius of 12 km (3 grid points). If the aggregated convective core has the major axis length of larger than 100 km, it is labelled as an MCS-core. Finally, the MCS-core is further aggregated stepwise until the aggregation exceeds 1) 96 km (24 grid boxes; also known as stratiform search radius)

or 2) the possible maximum radar-MCS boundary. Then this aggregated region defines the final radar-MCS. Note the choice of convective aggregation radius (12 km) is less than that of Haberlie and Ashley (2019; 24 km) for the consideration of different convective selection criteria (SL3D vs. composite reflectivity > 40 dBZ) used in the two algorithms. Other radar echoes within the IR-MCS shield are labeled as non-MCS related.

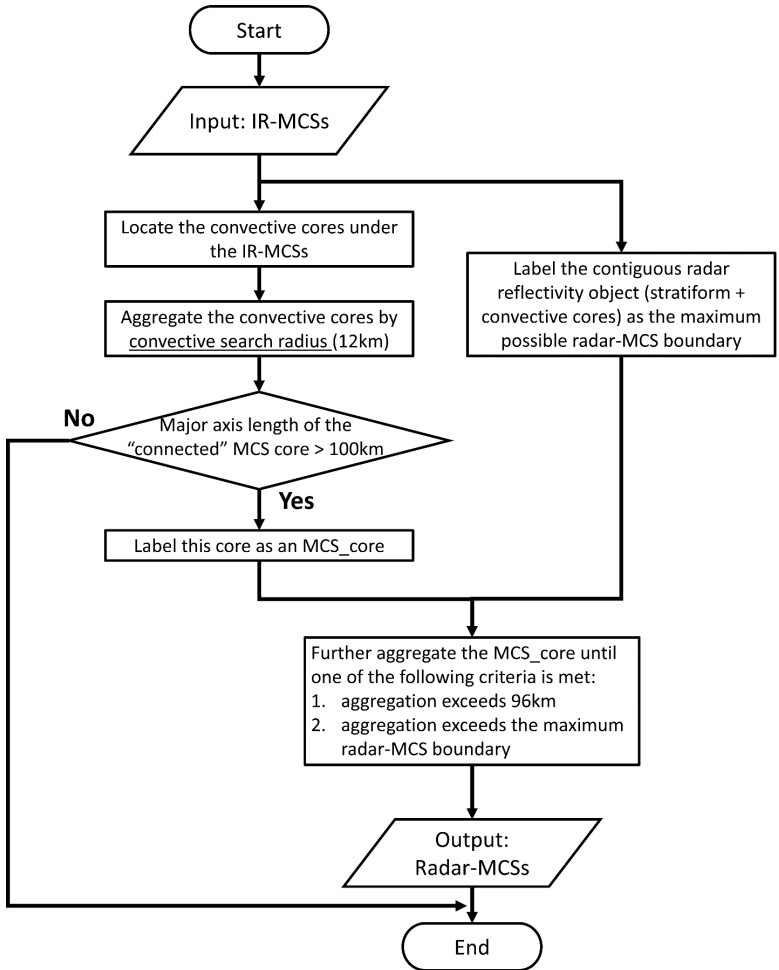

**Figure 1: Flowchart of MCS detection based on FLEXTRKR dataset.**



### 2.3 Mapping hail and tornado events to MCSs

The MCS dataset is available from 2004 to 2017 and the domain covers east of the Rocky Mountains (25°-48°N, 110°-70°W). Correspondingly the hail and tornado reports are collected for the same period over the same domain. The point-based hail and tornado reports are rasterized to the radar grid (4 km × 4 km) by summing all reports within each gridbox.

Because of higher uncertainty in reports of hailstone size less than 1 inch (2.54 cm, Jeong et al. 2020; 2021), only severe hail events with a size greater than or equal to 1 inch are included. For tornado, all events at EF0 and above are considered. Since there is a significant time difference between the MCS dataset (hourly) and the hazard reports (in minutes), it is critical to carefully examine how to match the two datasets temporally. For each hour in the MCS dataset, we tested different time intervals for assigning hail and tornado events to that hour: 1 hour (+/- 30 min), 30 min (+/- 15 min), 20 min (+/- 10 min),

and 10 min (+/- 5 min). Table 1 summarizes the total number of events and the missing rate (the percentage of the events occurring outside valid radar observations) for the different time intervals. For both hail and tornadoes, their total numbers decrease proportionally with the reduced time interval, indicating the events are evenly distributed across the entire hour. Meanwhile, their missing rate decreases drastically from +/- 30 min to +/- 15 min, but further reducing the time intervals to +/- 10 min and +/- 5 min has limited improvements in decreasing the missing rate. Thus, the +/- 15 minutes interval is

chosen to map the hazard records to the radar grid in the temporal dimension, as it contains sufficient sample numbers as well as a relatively low missing rate.

**Table 1: The statistics of the total number of records and the missing rate using different filtering time intervals.**

| Interval | Hail | | Tornado | |
|---|---|---|---|---|
| | Count | Missing Rate | Count | Missing Rate |
| +/-30 min | 133831 | 26.1% | 15222 | 25.2% |
| +/-15 min | 72753 | 18.2% | 7966 | 20.0% |
| +/-10 min | 49186 | 16.1% | 5447 | 18.7% |
| +/-5 min | 27531 | 14.9% | 3026 | 18.4% |

After mapping the hazard reports to the radar data grid (4 km grid resolution), the hail and tornado events are defined as "MCS-related" when they occur within the radar-MCS areal coverage, indicating those events are generated from the large, contiguous, and long-lived MCSs. Conversely, "Non-MCS-related" hazard events are labelled when they do not overlap with the radar-MCS objects, which are hazard events possibly produced by the smaller and shorter-lived convective systems that do not meet the MCS criteria.





Fig. 2 shows four examples of MCSs overlaid by their concurrent hazard reports. To eliminate the false inclusion of hazard reports that are irrelevant to the MCS-core, hail and tornadoes within the radar-MCS boundary but outside the MCS-core are not counted as MCS-related and reports without valid radar coverage are also discarded. During the 21-26 May 2011 tornado outbreak sequence, one of the largest tornado outbreaks on record, an MCS with the core length of 107 km generated an EF0 tornado to the Northwest of Shawneetown, Missouri on 20:54 UTC, 23 May 2011 (Fig. 2a). Meanwhile, sever hail events
were produced by an adjacent MCS (core length of 101 km) over southwest Indiana. By applying the modified FLEXTRKR algorithm as detailed in Fig. 1, three radar-MCSs previously under the same IR-MCS shield are separated. Even with much smaller spatial coverage, their inclusion of hazard reports is evident, indicating the robustness of the algorithm in extracting radar-MCSs as well as associating them with embedded hazard reports.

In addition to the sporadic occurrences of hailstone with relatively weak severity, reports with a more organized spatial
distribution and higher magnitude also demonstrate a strong association with MCSs. During the 18 March 2013 tornado and large-scale wind event (Fig. 2b), a very large bowing convective line (core length of 227 km) developed along a quasi-stationary frontal boundary extending northeastward from northern Louisiana into north-central Alabama, which spawned widespread hail events up to 2.75 inches. The forced ascent along the southeastward propagation of a gust front associated with the convective line aided the development of a supercell embedded in the line, which subsequently produced an isolated
EF2 tornado over Meriwether and Pike County, Georgia. This is an example in which a radar-MCS overlaps an IR-MCS without further subdivision. In spite of the greatly narrowed spatial coverage, the occurrences of hazard align well with the radar-MCS enclosed in the IR-MCS, and such MCS-hazard association is valid disregard of the criteria of MCS selection.

Similar as the densely distributed MCS-related hail events, tornadogenesis associated with MCSs can also be clustered. As shown in Fig. 2c during the Memorial Day Weekend Tornadoes (23-25 May 2015), 3 tornadoes (EF0-1) were reported along
a northeastward-bowing squall line (core length of 107 km) by a north-side embedded comma head vortex over Travis and Williamson, Texas. The cluster of tornadoes coincides with a clearly defined linear MCS, which fortifies the validity of the algorithm and proves that more severe tornadoes can also be observed within MCSs. The severity of EF2 is apparently not the worst tornadogenesis that MCSs can bring, as shown in Fig. 2d, a EF4 tornado was spawned during the 'Super Tuesday Outbreak' (a total of 131 tornadoes with the death toll of 57) on 6 February 2008 in Clinton, Arkansas. This long-track
tornado first touched down as EF0, then it rapidly intensified to EF4 along with a long-lived supercell embedded in the pre-frontal squall line (core length of 299 km). The association of such devastating tornadoes with the linear MCS is not a novel discovery by this study, but was also confirmed by previous studies (e.g., Molthan et al., 2008).

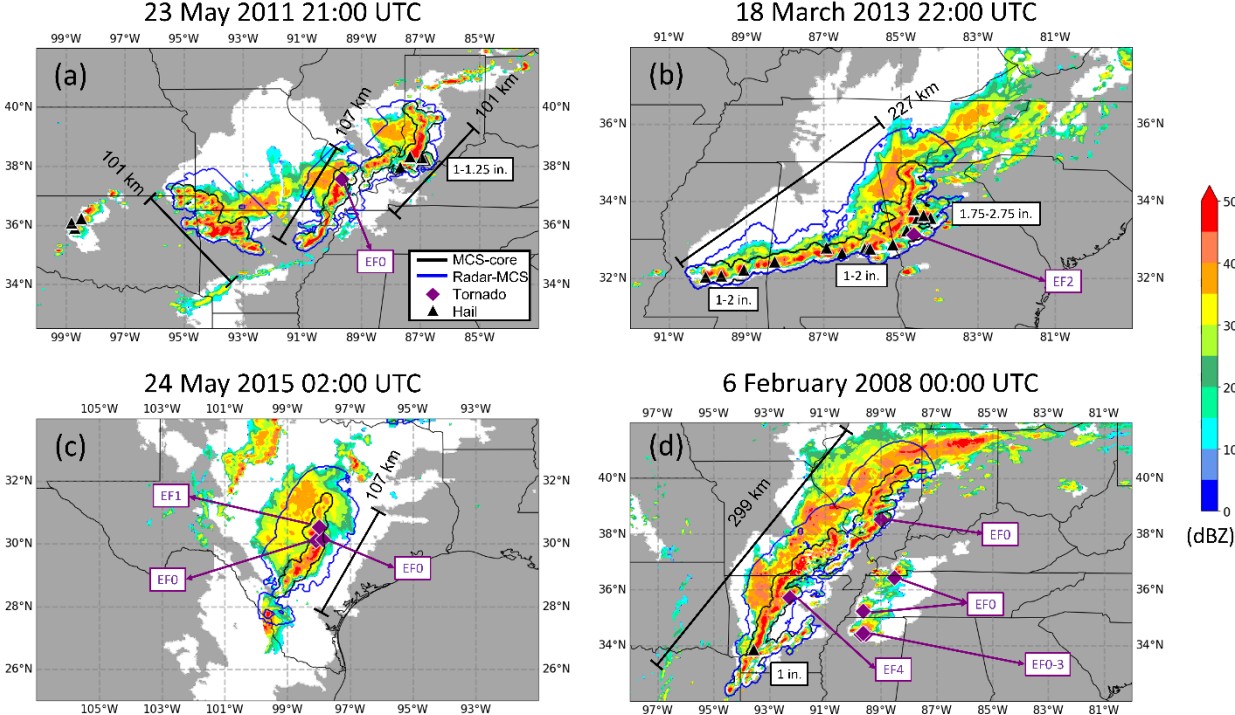

**Figure 2: Examples of mapping the instantaneous hail and tornado reports on the hourly MCS dataset for (a) 21:00 UTC, 23 May 2011, (b) 22:00 UTC, 18 March 2013, (c) 02:00 UTC, 24 May 2015, and (d) 00:00 UTC, 6 February 2008. The IR-MCS areal coverage is denoted by the white area. The blue outline represents the radar-MCS, and the black outline represents the MCS-cores. The 2-km altitude reflectivity is color contoured, and the reports of tornado (purple diamonds) and hail (black triangles) overlaid.**

## 3 Results

### 3.1 Spatial distribution of MCS-related hail and tornado events

Many previous studies have shown that the spatial distribution of severe hail events (hailstone size ≥ 1 inch) has notable regional and seasonal variation across the United States (e.g., Schaefer et al., 2004; Cintineo et al., 2012). However, those studies are commonly based on the climatology of the total number of reports without the discrimination between MCS and non-MCS. Fig. 3 decomposes the total counts (left column) into the MCS-related (middle column) and non-MCS-related portions (right column) for different seasons. The hazard reports rasterized to the 4 km resolution are further aggregated to 1°×1° in the figure for clarity. Annually, MCS-related hail events account for only 14% of the total, indicating MCSs contribute a limited portion of total hail occurrences, which is consistent with the findings in many previous studies that supercells dominate the hail production (e.g., more than 90% in Smith et al., 2012).

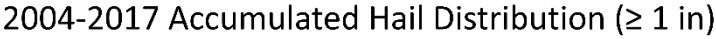

**Figure 3: Spatial distribution of hail occurrences per 1°×1° grid with reported hailstone size of 1 inch or greater. The counts of the total (a, d, g, and j), MCS-related portion (b, e, h, and k), and non-MCS-related portion (c, f, i, and l) are separated into the seasons of December-January-February (DJF), March-April-May (MAM), June-July-August (JJA), and September-October-November (SON), respectively. The numbers on the middle column panels denote the averaged percentages of MCS-related hail events relative to the total counts over the domain.**

Winter (DJF) has the least hail occurrences among all seasons; however, it corresponds to the highest MCS-related portion (23%), which agrees with the finding in Smith et al., (2012) that MCS is a typical mode of tornado-producing convection





nearly as frequent as the sum of other modes (Smith et al., 2012). According to Thompson et al. (2012), these MCS-related tornadoes were generated in the environment with the weakest buoyancy but the largest fixed-layer vertical shear, which was

less favorable for other modes of convection to produce tornadoes. Spatially, both total and MCS-related hail events are greatly confined to the southeast in this season (Fig. 3a-c), which reflects the dominance of cold air over the interior of the continent. In addition to the high static stability that prohibits the development of hail-generating severe thunderstorms, the low-level moisture supply is mostly confined to the southeast, where the narrow hail band is observed ranging from the east of Texas to the north of Alabama.

Spring corresponds to the maximum hail occurrences (Fig. 3d), because the strong baroclinic instability, in combination with the increased solar heating and northward expansion of moisture transport by the Great Plains low-level jet (GPLLJ), provides a favorable environment for the development and maintenance of organized convective systems (Maddox et al., 1979; Wang et al., 2019b). Therefore, the hot zone of hail occurrence shifts to the Southern Great Plains (SGP) over Oklahoma and the Arkansas Red River Basin. Although the MCS-related portion (Fig. 3e, 17%) is less than that of winter,

spring has the maximum number of MCS-related hail events among the four seasons. There are also significant amounts of hail events occurring in the Eastern U.S. (EUS), where the non-MCS-related fraction is much higher compared to the Central U.S. (CUS) because disorganized convective modes are common with local sea-breeze circulation (Smith et al., 2012), which also applies to summer (Figs. 3i).

In the summertime, although the baroclinic instability becomes the weakest (Holton and Hakim, 2004), the large solar

heating and the northwestward expanded GPLLJ along the sloping terrain (Burrows et al., 2019) promote local and small-scale convection. The hail center is further pushed to the Northern Great Plains (NGP) from eastern Colorado to Minnesota (Fig. 3g). In accordance with the changes in meteorological conditions, compared to spring, the hail events contributed by MCS decrease (Fig. 3h, 10%) and the non-MCS fraction increases (Fig. 3i, 90%). Besides the northwest centroid of high hail occurrence, another hail center is found in the EUS, where the non-MCS fraction is much larger than the MCS-related

fraction. The high fraction of non-MCS-related hail events in the East is due to the reduction of MCS and increase in non-MCS storms (Li et al. 2020), as the EUS summertime thunderstorms are mainly driven by daytime surface heating and land-sea circulation, characterized by smaller size and shorter duration compared to their counterparts in the CUS.

In the fall, the spatial pattern of hail activity is similar to spring as the large-scale environment is similar between the two seasons. However, the total occurrence number is greatly reduced by a factor of ~10 compared to spring. In spring, the lower

troposphere warms quickly but the high levels remain rather cold from winter, whereas in fall, the lower troposphere cools quickly while the high levels stay warm from the effects of summer. Thus, the instability is much weaker in fall compared to spring. However, the proportion of MCS-related vs. non-MCS-related is similar between spring (17%) and fall (14%). Thus, MCSs have a similar contribution to hail counts during the transitional seasons despite the contrasting total counts. Overall, the hail events occur mainly in the warm seasons (spring and summer); however, the maximum MCS-related percentage is

found in winter when the total hail occurrences reach the minimum.




For tornado, the distribution of total events shows similar spatial patterns as hail events (Fig. 4). This includes 1) winter has the least number of events and the occurrence is concentrated in the southeast, 2) spring has the maximum occurrences and the hot zone is located over the SGP, and 3) the high occurrence zone is shifted to NGP in summer. The only notable difference is that tornado spatial distribution in fall more resembles that in winter. where for hail, the spatial pattern in fall mimics its spring pattern.

## 2004-2017 Accumulated Tornado Distribution (≥ EF0)

Figure 4: Same as Fig. 3 except for tornadoes with enhanced Fujita Scale greater than or equal to 0.



Regarding the contribution from MCSs, the percentage of MCS-related tornadoes (27%) almost doubles that of hail events (14%) on the annual basis. Since the MCSs defined in this study contain a convective feature major axis length exceeding 100 km similar to QLCSs defined in many previous studies (e.g., Trapp et al., 2005; Ashley et al., 2019), we compare our tornado occurrence and attribution with those studies. A good agreement is found between our results and previous works. For example, based on a 3-year (1998-2000) and a 22-year (1996-2017) investigation, 18% and 21% of tornadoes were attributed to QLCSs respectively by Trapp et al. (2005) and Ashley et al. (2019). The higher MCS-related tornado occurrences in this study could result from the absent restriction of the aspect ratio (e.g., a QLCS major axis must be at least 3 times longer than its minor axis). Therefore, the inclusion of nonlinear MCSs would increase the MCS-related percentage. This would also explain the higher MCS-related hail occurrence (14%) than QLCS-related results (10%; Ashley et al., 2019). It is important to note that the choice of the convective aggregation radius (12 km; to connect the interspersed convective cores) and the stratiform search radius (96 km; to associate stratiform precipitation regions to the MCS convective cores) could affect the detection of MCS-core and its spatial coverage, which may subsequently impact the hazard statistics when matching the hazard reports to MCSs. Therefore, to quantify the uncertainties associated with the threshold selection, the sensitivity tests using 8 km for the convective aggregation radius (Figs. S1 and S2) and various stratiform search radii of 48 km (Fig. S3) and 192 km (Figs. S4) are conducted. With different combinations of the two criteria, the annual proportion varies from 8% to 17% for MCS-related hail events, and 17% to 32% for tornadoes, respectively.

### 3.2 Temporal variations of MCS-related hail and tornado events

Over 2004-2017, the annual total hail events vary significantly from 3,166 to 6,624, and the MCS-related portion fluctuates between 10% to 19% (Fig. 5a). Note that only the reports occurring with the +/- 15 min time window of the integral hours are considered in this study, therefore the total and MCS-related hazard counts are nearly half of those reported in other studies (e.g., Trapp et al.,2005; Ashley et al., 2019). The year 2011 stands out, when both the maximum total count of hail events and MCS-related portion peak. A similar large variation pattern is also found for tornado reports, where the annual count of tornadoes varies from 494 to 1,089, and the MCS-related portion varies from 18% to 45% (Fig. 5b). Same as hail, both total count and MCS-related percentage peak in 2011. As explained by Jeong et al (2020, 2021), the interannual variability of hail occurrence in Southern and Northern Great Plains is strongly modulated by El Niño/Southern Oscillation (ENSO) pattern, and 2011 is a strong La Niña year.  Other strong La Niña years of 2008 and 2017 also correspond to a relatively higher number of tornadoes and MCS-related fraction, indicating ENSO may modulate tornado occurrence during the winter and spring by altering the large-scale environments (Allen et al., 2015). The minimum total count occurs in 2007 (transition year from El Niño to La Niña) for hail but in 2013-2014 (weak La Niña condition) for tornadoes. Such inconsistency indicates that the total counts of different hazards could be modulated by different sets of large-scale drivers, which deserves future investigation. Through the examination of the correlation coefficients (r) in the interannual variability between the two types of hazards, the value for the total counts is as weak as 0.47. In comparison, their MCS-related portions are much more synchronized (MCS-related count: r = 0.88; MCS-related fraction: r = 0.62), indicating MCSs





generally produce both hail and tornadoes. Therefore, the study of environmental control over MCS-related hail and tornadoes could benefit from the same set of large-scale drivers.

The monthly distributions (Figs. 5c and d) show a consistent seasonal pattern with Figs. 3 and 4, where April-June (spring) has the maximum number of total and MCS-associated reports for both hazard types, and the minimum are found between

November and February (winter). The MCS-related portions for hail and tornado show a similar seasonal cycle: both peak in November-January and minimize in July-August. There is an apparent phase shift between the hazard counts (both total and MCS-related) and MCS-related percentages for both hail and tornado, which was also reported by Ashely et al. (2019) for QLMCS-related hazards (a subset of MCS). The seasonality of the hazard occurrence roughly follows the variation of atmospheric instability in combination with the latitudinal migration of large-scale synoptic features (i.e., fronts and

prefrontal troughs) associated with extratropical cyclones, producing the spring-peak-winter-valley pattern. However, both the MCS-related hazard fractions show the winter-peak-summer-valley pattern, which is approximately antiphase to the hazard counts. Such contrast may suggest that the winter environment that features the weakest buoyancy and the largest fixed-layer vertical shear is more favorable for hazard production in MCS than other modes of convection.

In summary, MCSs are most hazardous in spring (April-June) when the majority of MCS-related hail (68%) and tornado

(56%) events occur. This consistent seasonal pattern reveals that the two types of hazards, as well as their MCS-related portions, are more likely to occur during the transition season from mid-spring to early summer, as the surface and lower troposphere warms quickly but the mid-to-upper troposphere remains rather cold from winter. The high instability, together with the migratory extratropical cyclones and their fronts and prefrontal troughs (Whittaker and Horn, 1984; Lukens et al. 2018), provide a favorable environment for severe convective storms and the formation of MCSs (Johns, 1984; Guastini and

Bosart, 2016). The overall hazard occurrence is rare in winter, and a significant portion is from MCSs (up to 30% for hail) and (up to 51% for tornado).

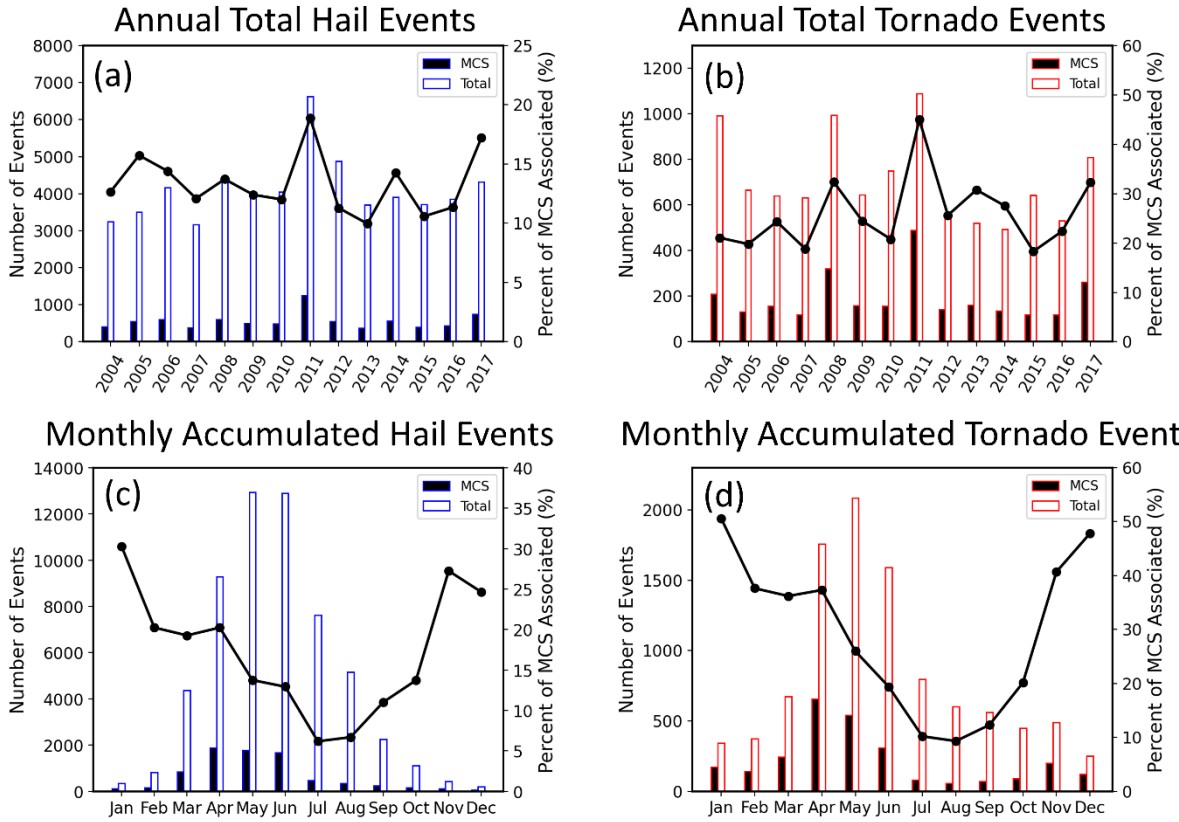

**Figure 5: The 2004-2017 annual variation of total (hollow bars) and MCS-related (filled bars) hazard events for (a) hail and (b) tornadoes, overlaid by MCS-related percentages in black lines. (c, d) similar as (a, b) but for the monthly statistics.**

The Lagrangian tracking database allows us to delve into the hazard evolution over the MCS lifecycle as shown in Fig. 6. As
defined by Feng et al. (2019), the initiation stage starts at the first hour when the MCS-related CCS is detected, followed by
the genesis stage when the major axis length of the convective core exceeds 100 km. As the convective core maintains its
size, the spatial expansion of the stratiform rain area defines the mature stage of MCS. Finally, when the length of the
convective core shrinks below the 100 km threshold, or the stratiform rain area is lower than the mean value throughout the
entire MCS life cycle, the system is classified as in the dissipation stage (Wang et al., 2019a).

On average, there are 454 MCSs per year (a minimum of 422 cases in 2005 and a maximum of 522 cases in 2015). 84/40 of
MCSs produce at least one severe hail event/tornado at certain stages of their lifecycle, and there are 29 MCSs with the
occurrence of both hail and tornado. By normalizing the hail-producing MCS lifespan from 0 to 1, we roughly define the
composite MCS lifecycle stages as follows based on the evolution of the convective and stratiform characteristics in the
MCS database (Fig. 17 in Feng et al. 2019): 0-0.1 represents the initiation stage, 0.1-0.4 for genesis, 0.4-0.8 for mature, and
0.8-1 for dissipation. The spaghetti plot in Fig. 6a shows the total count of MCS-related hail events each year. All years
show a consistent lifecycle variation in hail production, including the year of 2011 with abnormally high hail occurrences.


That is, there is a rapid increase in the initiation stage and peak during the genesis stage, then a gradual decrease towards the rest of the MCS life cycle (0.4-1). This pattern might be expected, as the most vigorous convective updraft commonly occurs
during the early stage of MCS which would be most hail productive. Whereas when the system becomes mature and starts the spatial expansion of stratiform cloud and precipitation, it would be less hail-productive, and finally there is negligible hail occurrence during the dissipation stage.

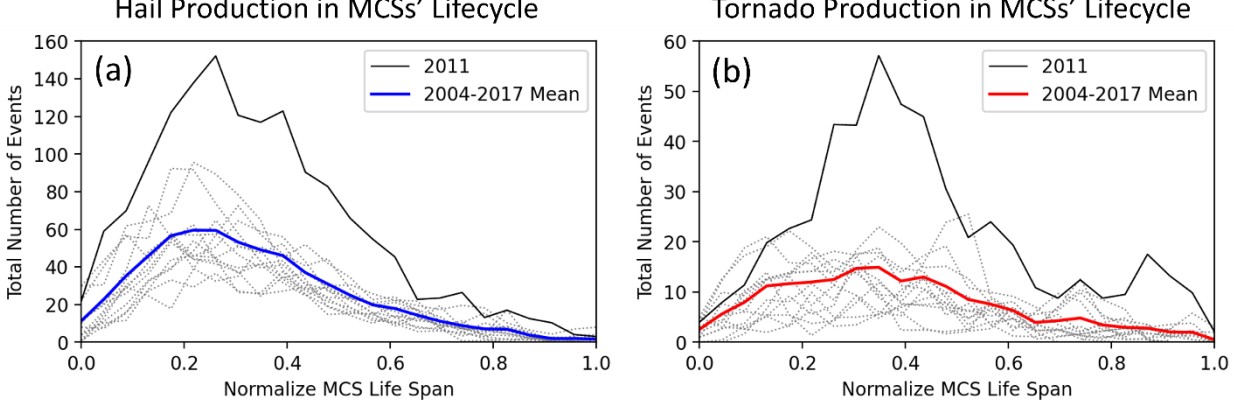

**Figure 6: (a) The total production of MCS-related hail events as a function of normalized MCSs' life span, where each dotted line**
**represents a year, except that 2011 is highlighted with a solid line, and all year mean is marked with blue. (b) is similar as (a) but for tornadoes.**

The tornado distribution along the MCS lifecycle is similar to that of hail; however, the average peak time is delayed from 0.25 to 0.35. Also, tornadoes can still occur near the end of the MCS life span (0.9-1.0). The difference in the patterns of hail and tornado occurrence along the MCS lifecycle echoes the different mechanisms for the production of the two types of
hazards. Hail production is most active in the upscale growth stage of MCSs, as the strongest updrafts, CAPE, and deep wind shear which affect hail formation and growth (Allen et al., 2020) are associated with young and vigorous convection. Tornadoes occur throughout much of the MCS lifecycle with a milder variation with different stages. This might not be surprising, as tornadoes can form along the outflow boundary produced by MCSs (Knupp et al., 2014; Markowski et al., 1998; Lee and Wilhelmson, 1997), which is more related to mesoscale downdrafts, precipitation, and evaporative cooling
rather than convective portion of MCSs. In addition to the direct tornadogenesis by convective cores embedded in parent MCSs, the interactions between supercells and the pre-existing, low-level thermal boundaries generated by MCSs were shown to augment the ambient horizontal vorticity (Markowski et al., 1998), which could facilitate the formation of mesocyclones and the subsequent tornadogenesis. For example, 70% of significant tornadoes were reported within 30 km of such boundaries during the Verifications of the Origins of Rotation in Tornadoes Experiment (VORTEX, Markowski et al.,
350    1998).



### 3.3 Severity characterization of MCS-related hail and tornado

As revealed in previous studies (e.g., Anderson-Frey et al., 2019), the spatiotemporal characteristics of hazard could vary dramatically in the spectrum of severities. However, the severity characterization of MCS-related hail and tornado has not been investigated. Fig. 7a presents the histograms of the 2004-2017 total number of hail reports as a function of hailstone size, overlaid by the MCS-related fraction. First, the number of events decreases by about an order of magnitude from severe hail (52,353) to significant severe hail (5,121), which is expected because larger hailstones are rarer compared to smaller ones. However, the MCS-related percentages are similar (13.7% for severe hail and 13.1% between the two hail categories, meaning that MCSs are not selective for producing a certain size of hail. For tornadoes, interestingly, the MCS-related fraction increases nearly monotonically from 20% at EF0 (1105 out of 5455) to 40% at EF4 and above (25 out of 63). As revealed by previous studies (Maddox et al. 1980; Markowski et al. 1998; Rasmussen et al. 2000; Wurman et al. 2007), a considerable number of tornadoes occurred near low-level boundaries (e.g., synoptic-scale fronts, or outflow boundaries produced by existing convection), which may be associated with the interactions between outflow boundaries of MCSs and emerging new convective cells in the vicinity. Such configuration has been reported when supercells move along or across those boundaries due to the enhanced low-level wind shear (Schultz et al., 2014), and many significant tornadoes with devastating outbreaks formed near low-level boundaries not associated with the forward or rear flank downdrafts of supercells (Markowski et al., 1998; Knupp et al., 2014). It has been hypothesized that these preexisting boundaries promote the production of tornadic low-level mesocyclones by augmenting the ambient horizontal vorticity (Markowski et al., 1998), however the boundary-supercell interaction and the quantification of how it would affect tornadogenesis warrants further observational and modeling studies.

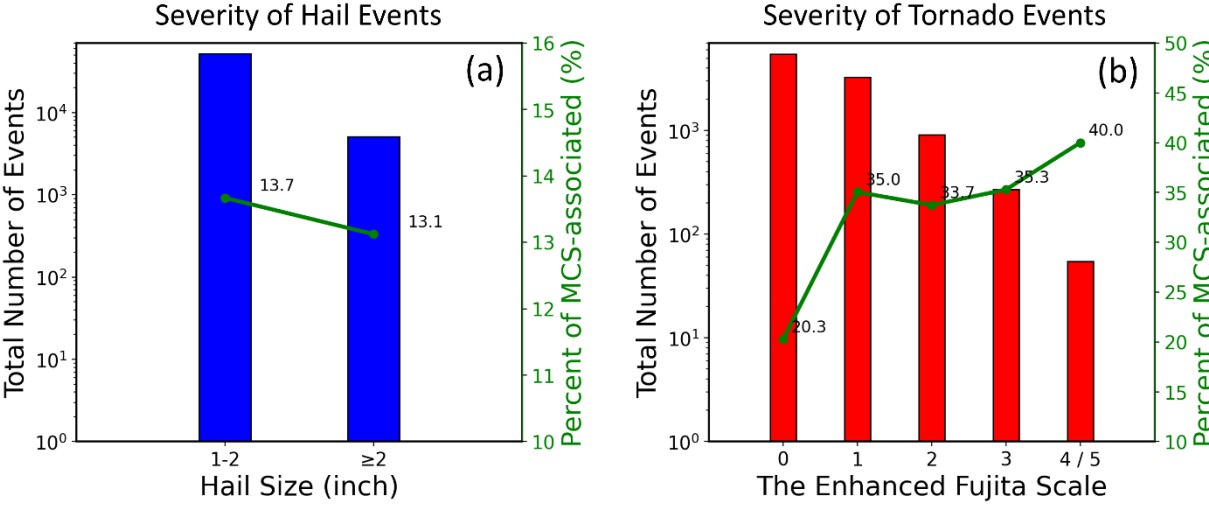

**Figure 7: (a) Dependence of the total number of hail events as a function of hailstone size overlaid by their fractions of MCS-related events in green lines. (b) Same as (a) except for tornadoes.**



A closer examination of the EF5 tornadoes and their associated MCSs are shown in Fig. S5, which all occurred on 27 April
2011 during the catastrophic tornado outbreak. It is important to note that the systems that generated the EF5 tornadoes in
375 the afternoon were classified as supercells (Kunupp et al., 2014) because of their cellular signatures in the low-level radar
observations. However, their reflectivity fields (> 20 dBZ) were connected at the upper level because of their proximity to
each other. As a result, the cores of those supercells were considered as parts of an MCS-core using the aggregation radius of
12 km. By applying a stricter convective aggregation radius of 8 km, the catastrophic EF5 tornadoes are no longer MCS-
related (Fig. S6b), and the MCS-related fraction drops to zero. The exclusion of the EF5 category does not change the
increasing trend of MCS-related fraction with tornado severity (Fig. S6b), indicating the robustness of this trend that is not
influenced by the choice of convective aggregation radius.

It is noteworthy that the genesis of EF5 tornadoes was strongly influenced by a low-level thermal boundary laid by an
MCS/QLCS that prevailed in the early morning and midday on 27 April 2011 (Knupp et al., 2014). Disregarding the EF5
tornadoes, the preceding MCS itself was impressively tornado-prolific, which spawned 76 tornadoes including 5 that reached
EF3 severity (Knupp et al., 2014). To eliminate the effect of outliers brought by the abnormally high occurrence of hazard
events in 2011 (Fig. 6), the year 2011 is excluded from the statistics (Fig. S7). Compared to Fig. 7, although all the MCS-
related fractions at different severity levels decrease, the trends of the two MCS-related fractions remain consistent.

In summary, we find an increasing contribution of MCS towards higher tornado severity that has not been shown in previous
studies.

**4 Conclusions and Discussion**

Quantifying the relationships of hail and tornado events with MCSs is of great importance to understanding the potential of
MCSs in hazardous weather production in addition to their roles in regulating the hydrological cycle. By carefully matching
the hazard report data from NCEI to a satellite-radar combined MCS tracking database spatially and temporally, we explore
the characteristics of hails and tornadoes and particularly quantify their relationships with MCSs based on a recent 14-year
period (2004-2017).

We find that 8-17% of hail events and 17-32% of tornadoes over the east of the U.S. Rocky Mountains are MCS-related,
which are slightly higher than the results from previous studies (e.g., 10% of QLCS-related hail occurrence in Ashley et al.,
2019; 18% to 21% of QLCS-related tornado occurrence in Trapp et al., 2005 and Ashley et al., 2019). The difference is
possible because a broader spectrum of MCSs is considered in this study besides the QLCSs, as about 60% of MCSs are
found to be non-linear (Cui et al. 2021). Interestingly, the MCS-related percentage for tornadoes is almost doubled relative to
hail, which may be related to the strong and organized outflow boundaries produced by MCSs that enhance low-level wind
shear and favor tornado formation.

Seasonally, the total count of MCS-related hail events peaks in spring over the SGP, followed by summer, when the MCS-
related hot zone shifts to the NGP. Although the total occurrence of MCS-related hail events in winter and fall is low, the





MCS-associated fraction reaches maximum in winter. Similar seasonality is also observed for tornadoes, and the monthly variation in MCS-related count of events and fraction is almost identical between the two types of hazards. The majority of MCS-related hail (68%) and tornado (56%) events occur in spring (April-June). Although the overall hazard occurrence is rare in winter, a notable fraction is from MCSs (up to 30% for hail and up to 51% for tornadoes). The consistent seasonal patterns show that the production of both types of hazards is highly concentrated during the period from mid-spring to early

summer, when the migratory extratropical cyclones and the associated synoptical features favor MCS formation (Song et al. 2019) and produce hazards (Johns, 1984; Guastini and Bosart, 2016) in an environment with high instability and baroclinicity (Whittaker and Horn, 1984; Lukens et al. 2018).

Hail and tornado demonstrate different patterns of occurrence during the MCS lifecycle. Hail production is most active during MCS upscale growth stage, as the strongest updrafts, CAPE, and deep wind shear promote the growth of young and

vigorous convection. Hail occurrence drastically decreases after MCSs enter the mature stage with the spatial expansion of the stratiform rain area. Finally, the decreasing trend persists toward the dissipation stage when both convective and stratiform rain areas shrink to minima. In contrast, the MCS-related tornado occurrence has a delayed peak time and decreases more slowly during the MCS mature stage compared to that of hail, suggesting the distinct mechanisms for the production of the two types of hazards in MCSs. MCSs have shown the production of tornadic low-level mesocyclones by

augmenting the ambient horizontal vorticity. Therefore, tornadogenesis is most frequent after the strongest convective development during the early growth stage and persists through the MCS mature stage.

Regarding the severity distribution, both the total and MCS-related number of events decreases as the severity increases for both hail and tornado events. However, the trends of MCS-related fraction with severity remarkably differ between the two types of hazards. For hail events, MCS-related fractions are similar between severe hail and significant severe hail (~13%).

In contrast, tornadoes with a larger EF scale have a higher chance to be associated with MCSs, which may be associated with the interactions between MCS outflow boundaries and emerging new convective cells due to enhanced low-level wind shear (Schultz et al., 2014).

By collocating an MCS tracking database with the hazard reports, this study shows that MCSs play a non-negligible role in the production of hail and tornado events, in contrast to previous works focusing on a subset of MCSs such as QLCS. More

importantly, the increasing contribution of MCS towards higher tornado severity has not been statistically revealed before. Given the fact that there are numerous hazard events occurring in the vicinity of MCSs, future work should focus on the dependence and interactions between these hazard events and the outflow boundaries produced by MCSs.

**Acknowledgements**

We acknowledge the support of the Department of Energy (DOE) Office of Science Early Career Award Program (70017) at

PNNL. PNNL is operated for the U.S. DOE by Battelle Memorial Institute under contract DE-AC05-76RL01830. This research used resources of the National Energy Research Scientific Computing Center (NERSC), a U.S. Department of



Energy Office of Science User Facility operated under contract DE-AC02-05CH11231. J. W. is supported by the Ministry of Education, Singapore, under its Academic Research Fund Tier 1 (RG74/22). The MCS dataset is obtained from the DOE Atmospheric Radiation Measurement (ARM) data archive (https://doi.org/10.5439/1571643). The hazard reports are
obtained from the National Oceanic and Atmospheric Administration (NOAA) National Centers for Environmental Information (NCEI) Storm Events Database (https://www.ncdc.noaa.gov/stormevents/ftp.jsp). The authors are grateful to Dr. James N. Marquis at PNNL for his insightful comments and suggestions.

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
