# Peer review of "Climatological occurrences of hail and tornado associated with mesoscale convective systems in the United States"

_Natural Hazards and Earth System Sciences, 2023_

## Author Comment (AC1)

Reviewer #1

**Summary:** The authors use collocated ground based radar MCS features and geostationary satellite tracked MCSs to follow MCSs over the United States throughout their lifetimes and define their life stage. Collocated observations of hail and tornadoes are collocated to each MCS to study the climatological occurrence of MCS produced severe hail and tornadoes on a monthly and per lifecycle stage basis. These are also compared with observed hail and tornadoes from all events to determine the percent contribution to the occurrence of these hazards by MCSs. The paper is well written and thorough, and, for the most part, logically organized. It would benefit from clarification of a few points and additional discussion of the methods used. I therefore recommend minor revisions.

We thank the reviewer for recognizing our efforts and providing helpful suggestions and comments. Our point-by-point responses are provided as follows.

**General Comments:**

1. The methods section is lacking in a few key details, mostly descriptions of the datasets. This section should be expanded to include more technical details about how the datasets used are produced and how the derived quantities are defined. Specific examples are provided in the Specific Comments section.

Technical details have been added to section 2 as suggested. Figure 1 has been modified accordingly.

[Figure]

2. The introduction feels a bit jumbled and tends to jump from discussing hail to tornadoes and back within the same paragraph. It would read more smoothly and thus make the points of the introduction better if it were reorganized to discuss the findings of previous studies about hail and tornadoes separately, rather than concurrently.

The introduction has been modified to discuss hail and tornadoes separately.

Hail can be produced by a variety of convective organizations (Trapp et al. 2005; Gallus et al. 2008; Smith et al. 2012), commonly associated with supercell thunderstorms (Moller et al., 1994; Prein and Holland, 2018), but is also found in multicellular storms (Nelson 1987; Kennedy and Detwiler, 2003), pulse storms (isolate, short-lived thunderstorms forming in a weakly forced environment; Miller and Mote, 2017), and mesoscale convective systems (MCSs; Houze, 2004). As a result, supercells are typically considered the most prolific producers of hail relative to other modes of convection, such as ordinary cells, multi-cells, MCSs, and unclassifiable types of convection (Duda and Gallus, 2010). The environments that support hail-producing storms show strong and persistent updrafts (Blair et al., 2017), large convective available potential energy (CAPE), 0-6 km bulk wind shear (Craven et al., 2004), as well

as 0-3 km storm-relative helicity (Prein and Holland, 2018), which are also observed within MCSs. Past studies on the relationship of hail with MCSs focused on quasi-linear convective systems (QLCSs; a subset of MCSs). Derived from a 22-year (1996-2017) observational dataset, 10% of severe hail events (with hailstone diameter greater than or equal to 1 inch) were attributed to QLCSs over the entire contiguous United States (CONUS) by Ashley et al. (2019). Using data from a shorter period (2003-2011), Smith et al. (2012) reported that 0.8% of significant severe hail events (with hailstone diameter greater than or equal to 2 inches) were produced by QLCSs.

Compared to the formation of hail, tornadogenesis may have different environmental conditions including high surface moisture supply and directional wind shear in the lowest 1-km (Rasmussen and Blanchard, 1998; Rasmussen, 2003), which are strongly associated with supercells (Smith et al., 2012; Schumacher and Rasmussen, 2020). As a result, Supercells are also known to be the most significant producers of tornadoes compared to other forms of convection, accounting for over 90% of tornado deaths in the contiguous United States (CONUS; Schoen and Ashley 2011; Brotzge et al. 2013). However, environments conducive to tornadogenesis are not exclusively limited to the isolated supercells. Some MCSs might produce tornados as revealed by observations (Nielsen and Schumacher, 2020) and numerical simulations (Nielsen and Schumacher, 2018). For example, a moist boundary layer with large, low-level vertical shear may induce supercell-like rotation within MCSs (Schumacher and Rasmussen, 2020). Past studies have attributed 13.8% to 21% of all tornadoes to QLCSs (Ashley et al., 2019; Smith et al., 2012). For significant tornadoes (with the Enhanced Fujita scale [EF] rating of 2-5), 20% of those events were linked to QLCSs from 2000 to 2008 (Grams et al., 2012). For the most extreme case, a QLCS has been reported for the unprecedented production of 76 tornadoes on 27 April 2011 (Knupp et al., 2014).

MCSs are considered the largest form of convective clouds (Houze, 2004) with spatial scales of 100s km and a lifespan of several hours to beyond a day (Houze, 2015; Feng et al., 2018, 2019). MCSs are commonly observed in many regions of the Earth (Wang et al., 2019a; Houze et al., 2019; Feng et al. 2021) and significantly impact precipitation, radiative forcing, and general circulation (Houze, 2018). Hazard-producing supercells, multi-cells and pulse storms can grow upscale and merge with nearby storms into an MCS. In addition, new quasi-linear or clustered convective features can be generated near pre-existing boundaries such as fronts or outflow boundaries, convectively generated gravity waves, or mesoscale convective vortices (Schumacher, 2009) within MCSs, and subsequently produce hail and tornadoes.

**Specific Comments:**

1. Line 27: The references of Moller et al 1994 and Nelson 1987 are quite old. It would be better to also include some newer references.

Newer references of Prein and Holland, 2018 and Kennedy and Detwiler, 2003 have been added.

Prein, A. F., and Holland, G. J.: Global estimates of damaging hail hazard. Wea. Climate Extremes, 22, 10–23, https://doi.org/10.1016/j.wace.2018.10.004, 2018.

Kennedy, P. C., and Detwiler, A. G.: A case study of the origin of hail in a multicell thunderstorm using in situ aircraft and polarimetric radar data. J. Appl. Meteor., 42, 1679–1690, doi:10.1175/1520-0450(2003)042<1679:ACSOTO>2.0.CO;2, 2003.

2. Line 86 "variety of sources": Please include some examples. Trained spotters? Automated stations?

Examples are provided as 'a variety of sources including local National Weather Service (NWS) services, public storm spotter networks, media reports, etc.'

3. Line 88 "Both hail and tornado reports are treated as point events…": This is an example of a definition that needs further clarification. If the tornado strengthens, or the hail size increases, do you use the maximum or the initial strength/size? If the maximum, is the time and location still counted as the start time?

Clarification has been added as 'Note that the data do not account for the temporal evolution of the hail and tornado events (i.e., the location and magnitude of the event was reported at its maximum intensity [National Weather Service, 2021]). An MCS could have multiple hail and tornado events and each event is distinguished.'

4. Line 88: How does the hail/tornado database determine if reports (especially hail) are continuous or if the hail stopped and started again? I.e. are there MCSs that have multiple hail point events associated with them and if so, is each event distinguished?

As clarified above, temporal evolution of hail and tornado events are not considered with the SPC reports. Yes, an MCS could have multiple hail and tornado events and each event is distinguished. This is also clarified now.

5. Line 96 "poorly identified": Consider rewording this statement. This makes it sound like MCSs are rare and hard to detect, which is not the case.

This sentence has been rewritten as 'there is a lack of an agreement regarding the specific observed variables and thresholds employed in the identification of MCSs.'

6. Line 106 "multiple properties": Please provide examples if you want to keep this amount of detail (see next comment).

The details of HA2018a/b have been removed.

7. Line 124 "convective cores defined by…": Why is the HA2018 method described in such detail if you are using S2017? Please either justify or remove this description.

The details of HA2018a have been removed.

8. Line 135 "maximum radar-MCS boundary": What is this boundary?

The word 'maximum' is confusing and thus removed. Clarification has been added as 'are marked as the radar-MCS boundary (i.e., the object that encloses the contiguous reflectivity objects).'

9. Line 147 "MCS dataset (hourly): Why is the MCS dataset limited to hourly resolution? Computational limitations? Or is one or more of the components of the dataset produced by others (please make this clearer if this is the case)?

It is the limitation of existing dataset used in FLEXTRKR. Both the 3D GridRad radar data and the Stage IV precipitation data used to create the MCS dataset were only available at hourly resolution at the time of their creation (Feng et al. 2019). Clarification has been added as 'since there is a significant time difference between the MCS dataset (hourly; the temporal resolution of FLEXTRKR product that this study is based on) and the hazard reports (in minutes), it is critical to carefully examine how to match the two datasets temporally.'

10. Line 153 "missing rate": It's not clear what you mean by this or why it would decrease with decreasing time.

The missing rate has been defined in the previous sentence.

The missing rate decreases with decreasing 'time interval within which the high-resolution hazard reports are matched to the hourly MCS dataset' not 'time'. Clarification has been added.

11. Line 166 "within the radar-MCS boundary but outside the MCS-core": Are these events related to the temporal offset of the hail/tornado and radar observations? I.e the hail or tornado was produced by the core, but the core is moving rapidly enough to not be over this location at the radar-MCS timestamp?

We thank the reviewer for pointing this out. Clarification has been added as suggested. 'It is important to note that the temporal offset between hazard reports and radar observations is not taken into account in this classification. Hence, instances of hail and tornadoes produced by fast-moving convective cores at sub-hourly scale are labelled as non-MCS-related or invalid records at the radar-MCS timestamp. It is also possible that spatiotemporal inaccuracies in the hazard report data (e.g., human-report errors, Trapp et al. 2006; Allen and Tippett, 2015) may affect the matching of the hazard reports and the radar features.'

12. Line 167 "reports without valid radar coverage": Does the definition of radar-MCS and the subsequent collocation of the hazard reports to the radar-MCS not already remove these?

Yes, those reports without valid radar coverage are already removed. This sentence has been removed.

13. Lines 231-236: Latitude definitions would be helpful here to those not familiar with US geography.

Latitude definitions have been added as suggested. ''

14. Line 355-356 "severe hail to significant sever hail": Please define (or remind of the definition) the sizes for these two categories

Reminder of the definition has been added.

Eastern U.S. (EUS; 110°W to 85°W)

Central U.S. (CUS; 85°W to 70°W)

**Technical Corrections:**

1. Line 49 "rotate": rotation

Corrected as suggested.

2. Line 99 "Erroneous": erroneous

Corrected as suggested.

3. Line 169 "sever": severe

Corrected as suggested.

4. Line 323 "severe hail event/tornado at": severe hail event/tornado, respectively, at

Corrected as suggested.

5. Line 351 "hazard": hazards

Corrected as suggested.

---

## Author Comment (AC2)

Reviewer #2

**Summary**: This is a nice study that leverages a long-term record of objectively identified mesoscale convective systems (MCSs) and severe weather reports to identify the extent of hail and tornadoes linked to this storm type. It builds upon past studies on the topic, but is unique in its inclusion of a broader collection of MCS type (most prior studies focus solely on quasi-linear systems). Most of the findings are consistent with prior work, with one exception for the rate of attribution of tornadoes to MCSs with increasing rating/intensity (EF-0 to EF-5). The manuscript is generally well-written and includes appropriate detail on data and methods. The figures are well designed and readable, though the rainbow color ramp used in several of the radar and density figures is not friendly to readers with color-vision deficiency. I have a number of general and specific suggestions for revision, which I outline below.

We thank the reviewer for the valuable comments to improve the paper, particularly the suggestion of adopting the color scheme friendly to readers with color-vision deficiency. Our point-by-point responses are provided below.

**General Comments**

1. One opportunity that seems missed, but well within reach of the authors is an attribution study for severe wind reports. Wind is not acknowledged by the authors apart from a brief mention for one of the cases highlighted in Figure 2. I recommend the authors add results for severe wind to their study or at least provide sufficient justification for their exclusion from this analysis.

Wind hazards have been excluded from this study because of their more diverse causative mechanisms and much more common occurrence compared to hail and tornado. Different from the dominance of continental convective process in generating hail and tornadoes, wind hazards can result from other meteorological phenomena like hurricanes, microbursts, tight pressure gradients, strong frontal systems, etc. Their distinct characteristics, occurrence frequency, and research needs warrant separate analyses. Furthermore, when examining the data in figure R1, it is evident that both hail and tornadoes often occur simultaneously with intense surface gusty winds. These wind events overshadow the signals of hail and tornadoes when plotted together. Therefore, instead of analyzing three types of hazards all together, this study prefers to focus on hail and tornadoes, leaving wind hazard to follow-on studies.

[Figure]

**23 May 2011 21:00 UTC**

Figure R1. Figure 2a overlaid by wind reports.

Such justification has been added. 'In addition to hail and tornadoes, MCSs are also prolific in producing wind hazards. Different from the dominance of continental convective process in generating hail and tornadoes, wind hazards can result from other meteorological phenomena like hurricanes, microbursts, tight pressure gradients, strong frontal systems, etc. Their distinct characteristics, occurrence frequency, and research needs warrant separate follow-on studies.'

2. The discussion in lines 216-224 conflates documented characteristics of MCS tornado production with interpretation of the hail associations evaluated here. This discussion is confusing and the parallels do not appear to be appropriate to make because the hazard production, its seasonality, etc. are not entirely similar for hail and tornadoes. All discussion of demonstrated tornado linkages should be left to the discussion of the tornado results.

The conflated discussion of tornado has been moved to later discussion of tornado results.

3. The discussion in lines 382-389 and 400-402 is too speculative. The limitation that supercells are not identified in the analysis is an important consideration here. Because prior studies focused on MCS/QLCS severe weather attributions

have been based on mostly manual evaluation of events with specific avoidance of supercells, the differing result found here may be a direct result of the inclusion of many supercells in your MCS classification. With supercells included, the increase in low-level wind shear in the early evening hours as the GPLLJ is established is also important to tornado production. Thus, I recommend softening some of the speculation here (and perhaps elsewhere) and acknowledging more the potential impact of the inclusion of supercells in your analysis. More specification of the differences between your analysis and prior analyses will also be helpful to making stronger assertions.

Such description has been modified as suggested. 'It is noteworthy the genesis of EF4 and 5 tornadoes has been commonly associated with supercells in prior studies (e.g., Smith et al., 2012; Knupp et al., 2014), which were excluded in studies focused on MCS/QLCS severe weather attributions (Trapp et al., 2005; Ashley et al., 2019) based on mostly manual examination of radar imagery. However, by using the automated algorithm featured in this study, considerable fractions of EF4/5 tornadoes (25 out of 63) were found MCS-related because of the supercells embedded in MCSs. By manual examination of the 25 MCS-related tornadoes with severity of EF4 and 5, it is observed that their respective low-level radar reflectivity field exhibits distinct supercell structures embedded in the MCSs. These supercell structures, however, were not evident in the composite radar reflectivity data. As a result, these particular records were not considered in the analysis.'

4. The 27 April 2011 discussions would benefit from citing the recent two-part paper summarizing an in-depth analysis of multiscale aspects of the event: https://doi.org/10.1175/MWR-D-21-0013.1 & https://doi.org/10.1175/MWR-D-21-0014.1

The suggested references have been included.

**Specific Comments**

Lines 17-18: I do not understand what the authors are aiming to communicate with this sentence. Please revise for clarity – perhaps it needs two separate sentence describing findings for hail and tornadoes.

This sentence has been rewritten as 'As hailstone size increases, the fraction associated with MCS decreases, but there is an increasing trend for tornado severity from EF0 to EF3. Violent tornadoes at EF4/5 associated with MCSs were also observed, which are generated by supercells embedded within MCSs.'

Line 30: "supercell is" should be "supercells are"

Corrected as suggested.

Line 48: "such as moist" should be "such as a moist"

Corrected as suggested.

Line 49: "rotate" should be "rotation"

Corrected as suggested.

Line 56: "tornado" should be "tornadoes"

Corrected as suggested.

Line 60: "to the large-scale" should be "to large-scale"

Corrected as suggested.

Line 64: "variabilities" should be "variability"

Corrected as suggested.

Line 71: "tornado with" should be "tornadoes within"

Corrected as suggested.

Line 169: "sever" should be "severe"

Corrected the typo.

Line 187: "more severe tornadoes". These are all EF0/1. What do you mean by more severe?

This sentence has been removed.

Line 192: I recommend noting that such associations are statistically rare. Should cite Trapp et al. 2005 also (https://doi.org/10.1175/WAF-835.1).

Note has been added with suggested reference.

Line 225: "because the" should be "because of the"

Corrected as suggested.

Line 227: "provides" should be "providing"

Corrected as suggested.

Line 228: recommend revising "hot zone" to "maximum frequency"

Corrected as suggested.

Line 238: "MCS decrease" should be "MCSs decreases"

Corrected as suggested.

Line 240: "reduction of MCS" should be "reduced frequency of MCSs" (I think)

Corrected as suggested.

Line 253: recommend revising "hot zone" to "frequency maximum"

Corrected as suggested.

Line 352-353: I do not understand what this sentence is aiming to communicate. Please revise for clarity.

This sentence has been rewritten as 'Previous research has indicated that the spatiotemporal features of hazards can exhibit significant variations across different levels of severity (e.g., Anderson-Frey et al., 2019). It would be interesting to understand how the severity of hail and tornado is associated with MCS events.'

Line 353: "tornado" should be "tornadoes"

Corrected as suggested.

Line 394: "hails" should be "hail"

Corrected as suggested.

Line 396: delete "over the"

Corrected as suggested.

Line 413: "tornado" should be "tornadoes"

Corrected as suggested.

Line 430: "MCS" should be "MCSs"

Corrected as suggested.

Figures 2-4: the colors used for reflectivity/density in these figures are not easily discernable to readers who suffer from color-vision deficiency. Good alternatives are

those which are perceptually uniform or divergent. If using Python, there are some good options here: https://matplotlib.org/stable/tutorials/colors/colormaps.html. For radar reflectivity in particular, *Spectral* is a good choice.

Figures 2-4 and S1-S5 have been replotted using the color map of 'Spectral_r'. Figure 2 is attached below for demonstration purposes.

[Figure]

Figure 6: "Normalize" on the x-axes should be "Normalized"

The x-axes of Figure 6 have been modified as suggested.

References: several of the citations here are missing DOI numbers. Please double-check for DOIs (even on older articles) and add all that are available.

The missing DOIs have been added.

---

## Author Comment (AC3)

Andrew Berrington (Reviewer #3)

In terms of the percentage of MCS-related tornadoes rated EF2-EF5 in Figure 7 and particularly for violent tornadoes, it has been shown in a number of studies that linear convective systems are not responsible for the vast majority of these events (Trapp et al. 2005, Smith et al. 2012). To observe a 40% fraction of violent tornadoes from MCSs suggests that the separation/delineation/classification between supercells and MCSs needs to be adjusted in this work. In the paragraph mentioning 4/27/2011, where the correct supercellular classification of the afternoon activity by Knupp et al. 2014 is mentioned, a stricter criteria is imposed by the authors that subsequently removes the EF5s from the MCS climatology. Stricter criteria and further quality control, perhaps further involving low-level reflectivity, should be applied to separate tornado cases that are clearly the result of supercells and those with MCSs. While embedded supercells within MCSs can indeed produce very strong tornadoes (e.g. the morning QLCS of 4/27/2011, 2/29/2012 in Harrisburg IL, and 4/13/2020 in Estill SC), the occurrence of these is very infrequent compared to more discrete supercells producing tornadoes of this intensity. While the definition of 'MCS' may include larger complexes including individual supercell storms such as the Feb 2008 case mentioned by the authors – especially using upper level reflectivity or satellite proxies where "connection" of elements may occur – it should be clear that discrete or semi-discrete supercells are a separate storm mode and thus these tornadoes should be classified separately.

References:

Smith, B. T., R. L. Thompson, J. S. Grams, C. Broyles, and H. E. Brooks, 2012: Convective modes for significant severe thunderstorms in the contiguous United States. Part I: Storm classification and climatology. Wea. Forecasting, 27, 1114–1135, https://doi.org/10.1175/WAF-D-11-00115.1.

Trapp, R. J., S. A. Tessendorf, E. S. Godfrey, and H. E. Brooks, 2005: Tornadoes from squall lines and bow echoes. Part I: Climatological distribution. Wea. Forecasting, 20, 23–34, https://doi.org/10.1175/WAF-835.1.

We thank the reviewer for the suggestion.

For the discrete supercells, our algorithm excluded them, and this has been clarified now at Section 2.3, 'In this way, all the irrelevant radar echo objects (including supercells, cellular and multicellular systems) are excluded from this study.'

For the supercell embedded in the MCSs, our automated algorithm identified that a considerable fraction of EF4 and 5 tornadoes (25 out of 63) were MCS-related. By manual examination of the 25 MCS-related tornadoes with severity of EF4 and 5, it is observed that their respective low-level radar reflectivity field exhibits distinct supercell structures embedded in the MCSs, these supercell structures, however, were not

evident in the composite radar reflectivity data. As a result, these particular records were excluded from the statistics in our analysis.

Although the EF4/5 tornado-producing supercells embedded in MCSs were detected in the automated algorithm employed in this study, such condition was rare for tornadoes at EF3 and lower intensity based on our sampling results (the complete manual screening of all tornado events is apparently infeasible). This reinforces the robustness of the observed increasing trend in MCS's fractional contribution to tornadoes with increasing severity.

Figures 7, S6 and S7, as well as the related discussion have been modified accordingly to exclude the association of EF4/5 tornadoes with MCSs.

---

## Referee Report (RR1)

Review of "Climatological occurrences of hail and tornado associated with mesoscale convective systems in the United States"

The authors have done a nice job responding to review comments. I have only a few technical edits to suggest below.

**Specific Comments**

Line 43: "Supercell" should not be capitalized

Line 47: "tornados" should be "tornadoes"

Line 67: "over southern" should be "over the southern"

Line 115: recommend revising "including supercells" to "including discrete supercells" for emphasis

A recent study that provides a comparable but distinctive analysis of the attribution of severe hazards to storm type (including discrete, multicell, and MCS) over the U.S. is highly relevant to this work and would be worth citing throughout where appropriate. Notably, because the study includes an objective identification for supercells, it helps to provide some additional perspective to the attribution results for hail and tornadoes:

https://doi.org/10.1175/MWR-D-23-0017.1